# Macrophage Polarization in Atherosclerosis

**DOI:** 10.3390/genes13050756

**Published:** 2022-04-25

**Authors:** Sahar Eshghjoo, Da Mi Kim, Arul Jayaraman, Yuxiang Sun, Robert C. Alaniz

**Affiliations:** 1Huffington Center on Aging, Baylor College Medicine, Houston, TX 77030, USA; sahar.eshghjoo@bcm.edu; 2Department of Microbial Pathogenesis and Immunology, Texas A&M University Health Science Center, Bryan, TX 77807, USA; 3Department of Nutrition, Texas A&M University, College Station, TX 77843, USA; dmkim5322@tamu.edu; 4Artie McFerrin Department of Chemical Engineering, Texas A&M University, College Station, TX 77843, USA; arulj@mail.che.tamu.edu

**Keywords:** macrophage, atherosclerosis, innate immunity, polarization, immunometabolism

## Abstract

The implication of the heterogeneous spectrum of pro- and anti-inflammatory macrophages (Macs) has been an important area of investigation over the last decade. The polarization of Macs alters their functional phenotype in response to their surrounding microenvironment. Macs are the major immune cells implicated in the pathogenesis of atherosclerosis. A hallmark pathology of atherosclerosis is the accumulation of pro-inflammatory M1-like macrophages in coronary arteries induced by pro-atherogenic stimuli; these M1-like pro-inflammatory macrophages are incapable of digesting lipids, thus resulting in foam cell formation in the atherosclerotic plaques. Recent findings suggest that the progression and stability of atherosclerotic plaques are dependent on the quantity of infiltrated Macs, the polarization state of the Macs, and the ratios of different types of Mac populations. The polarization of Macs is defined by signature markers on the cell surface, as well as by factors in intracellular and intranuclear compartments. At the same time, pro- and anti-inflammatory polarized Macs also exhibit different gene expression patterns, with differential cellular characteristics in oxidative phosphorylation and glycolysis. Macs are reflective of different metabolic states and various types of diseases. In this review, we discuss the major differences between M1-like Macs and M2-like Macs, their associated metabolic pathways, and their roles in atherosclerosis.

## 1. Introduction

Macrophages (Macs) are one of the major cell types of the innate immune system. They regulate inflammation and clear infection through antigen presentation, polarization, and phagocytosis. Macs release cytokines to regulate other immune cells [1]. Mac phenotypes exhibit a broad spectrum and, depending on the signals they are exposed to, are polarized to differentially activated states [2]. Macs manifest unique metabolic variations in different disease conditions, and atherosclerotic plaques are a unique example of Mac polarization and Mac-induced pathology [2]. Polarized Macs exhibit altered glycolytic metabolism, mitochondrial oxidative phosphorylation (OXPHOS) and lipid metabolism, as well as amino acid metabolism [2]. 

Quiescent Macs are referred to as M0 Macs. Classically activated pro-inflammatory M1 Macs and alternatively activated anti-inflammatory M2 Macs are the two most studied phenotypes of Macs [3]. The polarization of M0 Macs toward an M1 phenotype can be achieved by induction of bacterial lipopolysaccharides (LPS) via the PI3K-AKT-mTOR-HIF1α (Hypoxia-inducible factor 1-alpha) signaling pathway and M1 polarization can be further enhanced by interferon-gamma (IFN-γ) [4]. On the other hand, the polarization of M2 Macs is activated by Interleukin (IL)-4 or IL-13 via the JAK-STAT, PPARs, AMPK, and/or transforming growth factor-β (TGF-β) pathways [4,5,6]. The cell surface and intracellular markers of M1 and M2 Macs show distinctive characteristics [7,8,9]. For example, CD38 and MCP-1 are commonly recognized as M1 polarization markers, while CD206 and Arg-1 are commonly recognized as M2 markers. In addition to surface markers, to define the phenotype and functional relevance of Macs, it is important to assess the functional outcomes of the polarized Macs by determining the types of cytokines secreted and their effects on the surrounding cells [7,10,11,12]. The cell surface markers and cytokine signatures of M1 and M2 Macs are illustrated in Figure 1. 

Two main energy production pathways in cells are OXPHOS and glycolysis. Based on their microenvironment, Macs can choose to use either of these pathways, switch from one pathway to the other, or use both pathways [13]. M1 Macs have increased glycolysis, decreased oxidative phosphorylation (OXPHOS), producing inflammatory cytokines under infectious and inflammatory disease conditions [3,4]. In contrast to M1 Macs, M2 Macs have increased levels of OXPHOS and exhibit anti-inflammatory properties [4,5,6,14]. 

## 2. Mac Polarization in Atherosclerotic Plaques

Local inflammatory responses in atherosclerosis activate different cells within the atherosclerotic lesion [15]. Local inflammatory responses in atherosclerosis activate different cells within the atherosclerotic lesion. Endothelial cells are activated by lipids and inflammatory mediators in the vessel wall. Modified and oxidized LDL that enters the vein wall stimulates monocytes, these engulfed monocytes then infiltrate into the arterial wall [15,16,17]. Increased levels of modified LDLs or oxidized LDLs cause the migration of a significant number of monocytes into the atherosclerotic plaque area beneath the endothelial cells in the arterial wall [16]. The inflammatory microenvironment of the lesion induces the monocytes to penetrate the arteries and differentiate into Macs. Then, Macs phagocytize the modified lipoproteins, transform into foam cells, and eventually form the atherosclerotic plaques [17]. Inflammatory Macs release pro-inflammatory cytokines and induce inflammation. Consequently, more monocytes are further recruited to the lesion area, and the accumulation of foam cells eventually leads to the formation of a necrotic core of chronic atherosclerosis [18]. Macs have an important role in the phagocytosis of necrotic cells in the plaques; pro- and anti-inflammatory Macs exacerbate or alleviate the disease, respectively [19]. The different polarization of Macs affects their proliferation and capability to recruit more monocytes which eventually alter the abundance and diversity of specific Mac cell populations in the plaques [20,21]. As shown in Figure 2, different subclasses of Macs have been identified in the plaques based on their surface markers, functions, and cytokine production [20,21,22].

Within atherosclerotic lesions, cholesterol crystals, LPS, pro-inflammatory cytokines, and oxidized LDLs are known to induce pro-inflammatory M1 Macs [23]. Pro-inflammatory M1 Macs, normally activated through toll-like receptor (TLR-4) or nuclear factor NF kappa B (NFκB) pathways [24], secrete pro-inflammatory cytokines such as tumor necrosis factor alpha (TNF-α), IL-1ß, IL-6, IL-12, and IL-23 [24]. M1 Macs are the major inflammatory Mac cell population in lipid cores [25,26]. 

Alternatively activated M2 Macs, polarized locally or migrating from the circulating pool, produce anti-inflammatory cytokines such as IL-10 and TGF-ß, and they have a high phagocytotic capability in destroying dead cells and debris [27]. There are three subtypes of M2 macrophages. M2a Macs have been shown to have roles in wound healing, angiogenesis, and atherosclerotic lesions; M2a Macs are activated by IL-4 and IL-13 cytokines [28,29]. In contrast, M2b subtype Macs are induced by immune complexes along with IL- 1ß and LPS, they specifically express high levels of TGF-ß but different from other M2 subtypes, producing inflammatory cytokines such as IL-1ß, IL-6, and TNF-α [30,31]. Finally, M2c Macs are activated by glucocorticoids and TGF- ß; they phagocytize debris and apoptotic cells [32]. 

Other than M1 and M2 phenotypes, other polarized Macs in the plaques have been observed such as M(Hb), Mhem, Mox, and M4 [21,31]. M(Hb) and Mhem Macs, like M2 Macs are anti-inflammatory and produce anti-inflammatory cytokines such as IL-10, preventing the progression of plaque formation [33]. Ox-phospholipids induce Mox Macs that express IL1-ß and cyclooxygenase 2 (COX-2) and are regulated by the TLR-2 dependent metabolic pathway. Mox macrophages are counted as 30% of the total macrophages in the progressive plaques [33]. M4 Macs are categorized as pro-inflammatory and pro-atherogenic Macs in atherosclerosis [33]. M4 Macs are polarized by CXC chemokine ligand 4 (Cxcl4) associated with reduced phagocytosis, producing inflammatory cytokines and molecules such as IL-6, TNF-α, and MMP-7 [31]. 

These polarized Mac subtypes regulate other Mac subpopulations to modify in their microenvironment, together inducing aggravation or regression of the plaques [21,31]. It is important to note that polarized Macs have plasticity and the capability to depolarize, switching their phenotype and function based on the microenvironment [21].

## 3. Lipid Metabolism in Macs to Promote Anti-Inflammatory Polarization

The most efficient pathway for producing ATP in the cells is fatty acid oxidation (FAO). For example, one palmitate molecule (FA contains 16 carbons) can produce 129 ATPs [31,34,35]. Excess cholesterol causes Mac-mediated foam cell formation; Macs uptake the lipoproteins from apoptotic cells and activate signaling pathways to reduce cholesterol in the cells [35]. Through scavenger receptors such as CD36, phagocytosis, and micropinocytosis, Macs uptake modified LDLs and VLDLs, and these lipids then are catabolized in the lysosome into FA and cholesterols [36]. In the endoplasmic reticulum (ER), free cholesterols are esterified and form cholesterol fatty acid esters. If not degraded or cleared, the lipid molecules accumulate in the cytosol as lipid droplets and shift the Mac towards foam cell formation [37]. Alternative pathways exist to export free cholesterols through the cell membrane [38]. Excessive cholesterol accumulation leads to increased expression of transcription factors such as retinoid x receptor (RXR) and liver x receptor (LXR) that upregulate the ABCG1 and ABCA1 expression [39]. These lipid transporters mediate the efflux of the cholesterol via intermediate pathways to form HDLs [40]. It has also been suggested that the possibility of aqueous diffusion and facilitated diffusion pathways are involved in cholesterol transport [41].

Degradation of the lipids to fatty acids and free cholesterol in the lysosome takes place through acid lipase enzymes in a process called lipolysis [42]. After efflux, cholesterol associates with LDLs, and the fatty acids enter the FAO process [43] where they are converted into fatty acid acyl-CoA and enter the mitochondria through an enzymatic process of carnitine palmitoyltransferase 1A (CPT1A) [44]. Once in the mitochondria, carnitine is removed by the CPT-2 enzyme, and β-oxidation of FA-acyl-CoA occurs [44,45]. FAO leads to increased production of acetyl-CoA through the citric acid cycle, increasing production of NADH and FADH that consequently produce ATP through the electron transport chain in the mitochondria [46].

On the other hand, Macs need lipids for proliferation and growth [43]. Fatty acid synthesis (FAS) is a metabolic pathway that generates FA in the cytoplasm using the metabolites from the Krebs cycle, pentose phosphate pathway, and glycolysis [47]. Moreover, mTOR activation induces FAS through the transcription factor sterol regulatory element-binding protein (SREBP) [48], while mTOR inhibition leads to the activation of autophagy and lipophagy that together reduce lipid accumulation in Macs [49].

FAO activation is one of the main metabolic pathways in M2 Macs [31]. IL-4 stimulated M2 Macs induce FAO which is dependent on PPARs and their co-activator peroxisome proliferator-activated receptor gamma co-activator 1-b (PGC-1b) to increase mitochondrial biogenesis [50]. The dependency of M2 Macs on FAO is still controversial. Some publications showed that inhibiting FAO in M2 Macs in humans and mice, beta-oxidation inhibitor etomoxir was not able to reduce M2 polarization in IL-4 polarized M2 Macs [51,52]. The results regarding M1 Macs suggested that when modified LDL and free FA uptake were increased, the expression of the scavenger receptor was upregulated, while lipolysis and FAO were reduced [52].

It is not well understood how diverse Mac phenotypes and their associated metabolic pathways are coupled with their phenotypes, meet their energy demands, or affect atherosclerotic plaque stability. However, the current literature supports the notion that M1 Macs accumulate more lipids compared to M2 Macs; M1 Macs induce inflammation and have less activated FAO, while M2 Macs have more active FAO and FA consumption [53,54].

## 4. Glycolysis Modulation in Macs to Promote Pro-Inflammatory Polarization

Macrophages that are activated by pathogen-associated molecular patterns (PAMP) via TLR or other pro-inflammatory factors are M1-like Macs, exhibiting increased glycolysis and decreased OXPHOS [52,54]. Glycolysis is an alternative energy-producing mechanism, it is a faster but a much less efficient way to produce ATP compared to OXPHOS, glycolysis produces 2 ATP, while OXPHOS produces 26 ATP [52]. Inhibition of glycolysis reduces M1 polarization of Macs. Reduced OXPHOS activation in pro-inflammatory Macs results in the accumulation of Krebs cycle metabolites such as malate, fumarate, citrate, and succinate [54]. Figure 3 illustrates the various intermediates used as precursors for anaerobic glycolysis in pro-inflammatory M1-like Macs [55].

Furthermore, increased glycolysis uncouples the mitochondrial electron transport chain from ATP synthesis, causing increased ROS production levels [56] which is one of the mechanisms that Macs use for bacterial killing [57]. mTOR, hypoxia-inducible factor 1-a (HIF-1α), and ubiquitous enzyme 6-phosphofructo-2-kinase/fructose-2,6-bisphosphatase (uPFK2) are known to be involved in the regulation of glycolysis [58]. The uPFK2 increases glycolysis by increasing the 6-phosphofructo-1-kinase (PFK-1) enzyme activity, and consequentially the production of more fructose 2,6-biphosphate in pro-inflammatory Macs [58]. There are also elevated levels of glucose uptake in M2-polarized Macs compared to naïve (M0) ones, but M1 Macs have the highest glycolysis activity (M1 > M2 > M0) [59]. As discussed above, to maintain their anti-inflammatory functions, M2 Macs use the Krebs cycle and OXPHOS pathways. However, other studies do suggest that M2 Macs can also use the FAO pathway to meet their anti-inflammatory energy needs [60], which is illustrated in Figure 4.

## 5. Models and Methods for Mac Polarization Study

In order to experimentally study Mac polarization and function, it is important to understand the cell type and cell origin [61,62]. For example, immortalized Mac cell lines and primary cells are fundamentally different cell types. In addition, human, mouse, or rat macrophages have certain unique and distinct characteristics. Mac cell lines are not very good candidates for polarization studies because of their immortal nature which exhibit an “activated” phenotype which may be misleading for polarization studies [62,63]. On the other hand, primary cells are the most physiologic source, and gene expression is closely associated with their polarization states and metabolic pathways [62,64]. 

Although murine peritoneal macrophages have been frequently used in macrophage studies for a long time, they are not the best cell types for polarization studies because they are often collected from the peritoneal cavity after elicitation/activation by sterile inflammatory agents (e.g., thioglycolate) and the collected peritoneal exudate contains heterogeneous cell types which may be affected by various polarization agents [65]. Murine bone marrow-derived Macs (BMDMs) are thought to be a better system for basic Mac polarization studies. As a highly enriched population of primary-derived cells, they are naïve and quiescent, and their gene expression and metabolic pathways are reflective of the polarization cues [66,67]. 

Given the variety of polarized Mac subtypes, identifying polarized macrophages with a set of markers rather than a single marker, also including cytokine section readout, will provide a more functionally relevant result in characterizing Mac phenotypes [68]. Transcriptomics (such as microarray) provide a wide range of transcript data regarding the polarization markers [69]. Real-time qPCR provides more sensitive data with a smaller number of cells [70]. Single-cell RNA-seq provides information regarding the wide spectrum of the cell population of the polarized Macs [71]. However, it needs to be noted that all aforementioned methods are gene-based which have the common limitation that the gene expression profile is not necessarily always correlated with protein levels/functional outcomes.

Protein expression studies are generally more functionally relevant than gene expression. ELISA, Luminex assays, and mass spectrometry techniques are some examples of protein-based detection of Mac polarization [72,73,74,75]. The flow cytometry technique is among the most reliable and commonly used techniques to assess phenotype and function, because of its cell-surface marker and cytokine labeling advantages [76]. In addition, metabolic assays using extracellular flux analyzers are attractive methods to evaluate the functional profiles of the Mac’s polarization of different cell populations [59]. 

We believe that the combination of methods mentioned above would also be of advantage for in-depth investigation of Mac heterogeneity. Better understanding of the Macs’ heterogeneity unique for specific disease stages of atherosclerosis would help to develop diagnosis methodologies and personalized treatments. For example, understanding the specific cell signatures that clearly differentiate the Mac phenotypes would help to determine the exact type of Macs in plaques during different stages of atherosclerosis, which would be very valuable in guiding better diagnosis and treatment.

## 6. Conclusions

Atherosclerosis is the leading cause of death in western countries. In the atherosclerosis disease model, studies regarding the role of Mac regulation and their polarization are very valuable in the prevention of foam cell formation and consequently control of atherosclerosis. Atherosclerotic plaque Macs resemble the polarized M1 phenotype, showing elevated inflammation and glycolytic metabolism. In contrast, M2 Macs prevent plaque formation, showing reduced inflammation and oxidative phosphorylation. Mechanisms that minimize Mac inflammation, increase lipid degradation, and prevent foam cell formation, are likely to decrease atherosclerosis progression. Future works are needed to further elucidate the mechanisms of actions by which different factors induce inflammatory or anti-inflammatory Macs in the context of foam cell formation. For example, microbiota has emerged as a key regulator of a variety of immune cell functions [77], including macrophages [66,76]. It would be important to investigate the role the microbiota plays in regulating macrophage polarization in general, and specifically in atherosclerosis development. 

Overall, this review underscores the importance of macrophage polarization, and its critical role in the pathogenesis of atherosclerosis. A better understanding of Mac infiltration, differentiation, polarization and phagocytosis would be extremely beneficial for the prevention and treatment of atherosclerosis.

## Figures and Tables

**Figure 1 genes-13-00756-f001:**
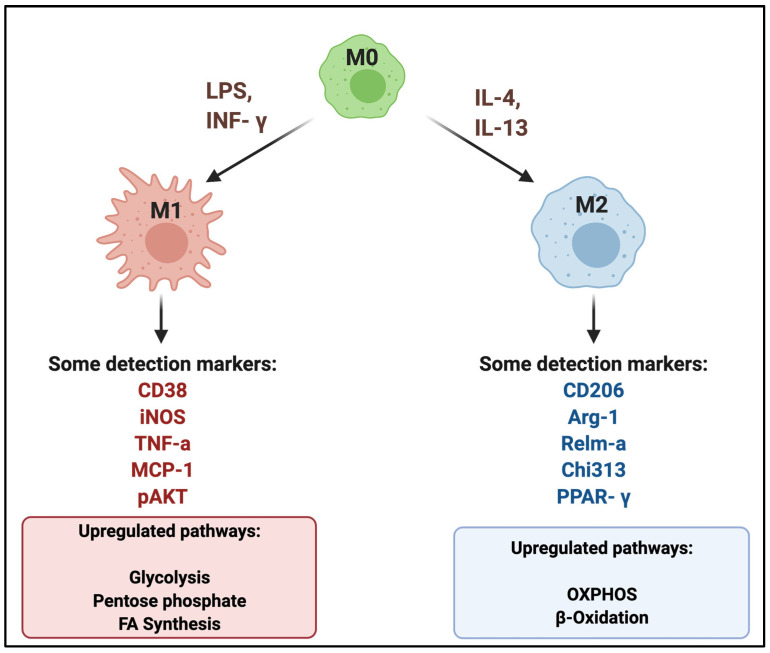
**Canonical M1 and M2 polarization of Macs.** Under a range of polarization signals, naïve Macs are polarized toward M1-like (LPS, IFN-γ skews) or M2-like (IL-4, IL-13 skews) Macs. There are several surfaces, intracellular or intranuclear markers for the detection of Mac phenotypes. Other than recognition markers, some specific pathways are upregulated or downregulated in M1 and M2 Macs. For example, glycolysis, pentose phosphate pathway, and fatty acid synthesis are upregulated in M1 Macs, while oxidative phosphorylation and β-oxidation pathways are upregulated in M2 Macs.

**Figure 2 genes-13-00756-f002:**
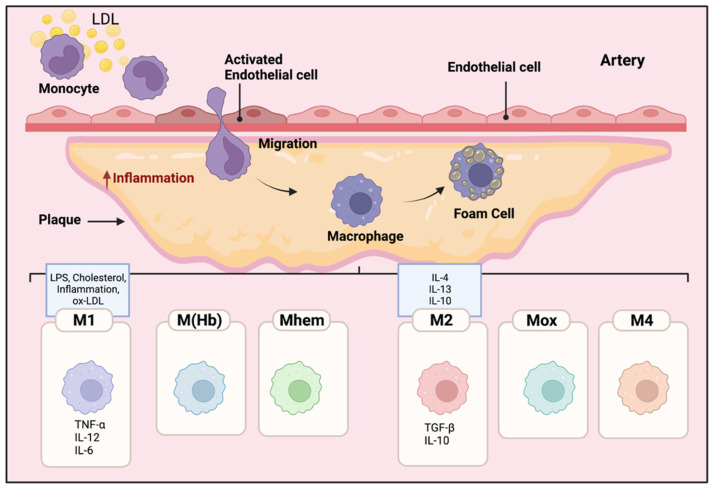
**Mac polarization in atherosclerotic plaques.** The microenvironment of the lesion affects the differentiation of monocytes to Macs. The Macs transform into foam cells and are retained in the plaques. Therefore, pro and anti-inflammatory Macs exacerbate or alleviate the disease, respectively. The polarization of Macs changes their functional phenotype in response to the signals in their microenvironment. Different subclasses of Macs have been identified in the plaques including M1, M(Hb), Mhem, M2, Mox, and M4. M1 Macs in atherosclerotic lesions can be stimulated by cholesterol crystals, LPS, pro-inflammatory cytokines, and oxidized LDL. They secrete pro-inflammatory cytokines such as TNF-α, IL-6, IL-12; in contrast, activated M2 Macs produce anti-inflammatory cytokines such as IL-10 and IL-4.

**Figure 3 genes-13-00756-f003:**
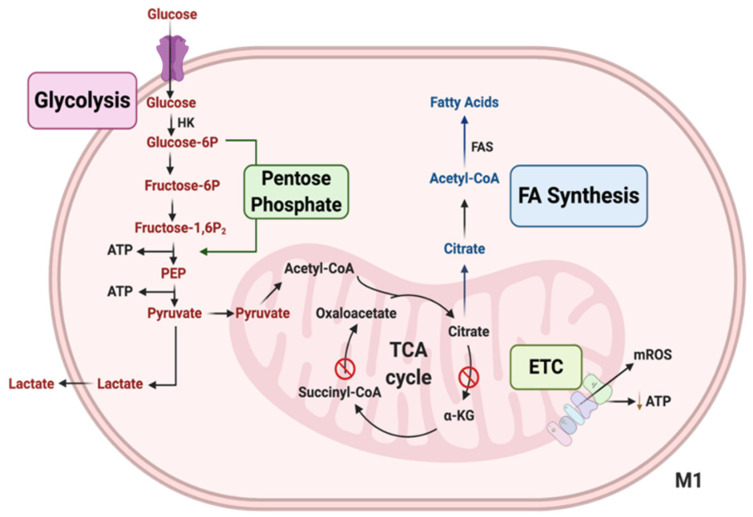
**Important metabolic pathways in M1 Macs.** M1 Macs have increased aerobic glycolysis, the process of converting glucose into lactate. An increased pentose phosphate pathway (PPP) is another characteristic of M1 Macs, which leads to the generation of the inflammatory mediators, nitric oxide (NO) and reactive oxygen species (ROS). M1 Macs have enhanced levels of succinate and citrate because the TCA cycle is partially inhibited. Furthermore, increased citrate leads to enhanced fatty acid synthesis (FAS) in M1 Macs. Reduced levels of electron transport chain (ETC) activity and production of mitochondrial ROS are also known hallmark features of M1 Macs.

**Figure 4 genes-13-00756-f004:**
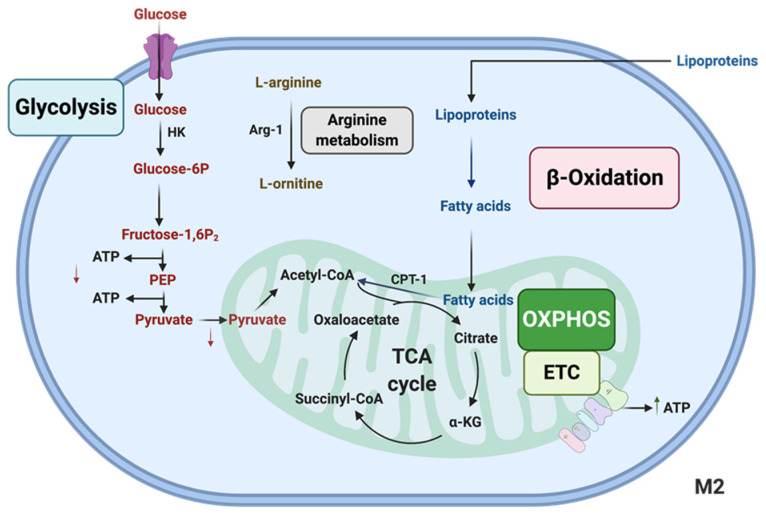
**Important metabolic pathways in M2 Macs**. M2 Macs have increased ATP production through electron transport chain (ETC) oxidative phosphorylation (OXPHOS). M2 Macs show lower levels of glycolysis, increased levels of fatty acid oxidation (β-Oxidation) and increased arginine metabolism.

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
