# Peer review of "Macrophage Polarization in Atherosclerosis"

_genes, 2022, doi:10.3390/genes13050756_

Round 1

Reviewer 1 Report

Sahar Eshghjoo et al. in this review have treated a very important subject “Macrophage polarization in atherosclerosis”. The review is interesting and discuses about macrophage polarization and phenotypic, functional and metabolic differences between M1 and M2-like macrophages. The role of different subclasses of macrophages in atherosclerosis is also treated.

Some remarques and suggestions are however required.

  1. Introduction:

The idea presented, line 54 to 56, does not seem to correspond to the cited reference [12; Gu, Q., et al., Adv Healthc Mater, 2017. 6(17)]. Can the authors verify this?

  1. Mac polarization in atherosclerotic plaques:

For more clarity, I would suggest to modify the first paragraph (line 73 to 77). The early stages of atherosclerosis should be discussed succinctly but in chronological order (i.e. activation of the endothelium, infiltration of LDL and monocytes, modification of LDL and differentiation of monocytes into macrophages...).

Figure2: At the top of the figure, the legend of LDLs and activated endothelial cells (in dark red) is missing. Also change "Athery" to "Arthery". M(Hb), Mhem, Mox and M4 must be cited in the legend.

Line 101 to 102, M1 Macs are the most abundant 101 Mac cell population in lipid cores [25]. I can't find this reference, can the authors verify its accuracy?

“TNF-a” and “HIF-1a” must be replaced by “TNF-α” and “HIF-1α” in the manuscript.

  1. Lipid metabolism in Macs to promote anti-inflammatory polarization

The last paragraph (line170 to 175) refers to the work of Garcia-Sabaté et al. which does not correspond to the reference [12]. This reference, also commented in the introduction, must be absolutely checked and corrected.

Author Response

Dear Reviewer,

We greatly appreciate your insightful comments of our manuscript, your constructive critiques have helped us to improve the manuscript tremendously. We have addressed all the comments of the reviewers. Please see revised manuscript and point-by-point responses in text below.

Yuxiang

  1. Introduction:

The idea presented, line 54 to 56, does not seem to correspond to the cited reference [12; Gu, Q., et al., Adv Healthc Mater, 2017. 6(17)]. Can the authors verify this?

Reply: We thank the reviewer for catching this oversight. The related reference and sentences have been deleted.

Mac polarization in atherosclerotic plaques:

For more clarity, I would suggest to modify the first paragraph (line 73 to 77). The early stages of atherosclerosis should be discussed succinctly but in chronological order (i.e. activation of the endothelium, infiltration of LDL and monocytes, modification of LDL and differentiation of monocytes into macrophages...).

Reply: The first paragraph (line 73-97) has been modified in the chronological order as the reviewer kindly suggested.

Figure2: At the top of the figure, the legend of LDLs and activated endothelial cells (in dark red) is missing. Also, we have changed "Athery" to "Arthery". M(Hb), Mhem, Mox and M4 must be cited in the legend.

Reply: The LDL and activated endothelial cells are now noted in the figure, and the artery typo is corrected. Macrophage sub-types are added in the legend.

Line 101 to 102, M1 Macs are the most abundant 101 Mac cell population in lipid cores [25]. I can't find this reference, can the authors verify its accuracy?

Reply: The “most abundant” has been revised to “major inflammatory” to better reflect the reference. Also, another new reference (number 26) has been added for better clarification.

“TNF-a” and “HIF-1a” must be replaced by “TNF-α” and “HIF-1α” in the manuscript.

Reply: All TNF-a and HIF1a have been changed to TNF-α and HIF-1α.

  1. Lipid metabolism in Macs to promote anti-inflammatory polarization

The last paragraph (line170 to 175) refers to the work of Garcia-Sabaté et al. which does not correspond to the reference [12]. This reference also commented in the introduction, must be absolutely checked and corrected.

Reply: Appreciate the reviewer helping us to identify this error, the paraph and the reference 12 have been deleted.

Reviewer 2 Report

The article is interesting, clearly structured, and well illustrated. The list of references contains sufficiently up-to-date sources. Minor comments:
1.    The first mention of macrophages in the text should be written in full
2.    In Fig. 2 there may be a typo in the word Athery. There are single other misprints in the text.
3. It would be useful to add more information about the involvement of different macrophage subtypes in inflammation and atherogenesis.
4.    More information on how understanding macrophage heterogeneity is useful for improving approaches to diagnosis/treatment of atherosclerosis would be recommended. May be directions for future research.

Author Response

Dear Reviewer,

We greatly appreciate your insightful comments of our manuscript, your constructive critiques have helped us to improve the manuscript tremendously. We have addressed all the comments of the reviewers. Please see revised manuscript and point-by-point responses in text below.

Yuxiang

The article is interesting, clearly structured, and well-illustrated. The list of references contains sufficiently up-to-date sources. Minor comments:
      1.    The first mention of macrophages in the text should be written in full

Reply: We have changed the first mac to macrophage (Mac).

      2. In Fig. 2 there may be a typo in the word Athery. There are single other misprints in the text.

Reply: The type in the figure has been corrected.

      3. It would be useful to add more information about the involvement of different macrophage subtypes in inflammation and atherogenesis.

Reply: We have revised the paragraph to provide more information about the involvement of other macrophage subtypes. The following paragraph has been added in lines 147-153.

      4. More information on how understanding macrophage heterogeneity is useful for improving approaches to diagnosis/treatment of atherosclerosis would be recommended. May be directions for future research.

Reply: Thank you for the suggestion. To better address macrophage heterogeneity, in addition to newly added section mentioned above (Lines 147-153), we have also added lines 339-345 to discuss the potential implication of macrophage heterogeneity in diagnosis/treatment of atherosclerosis.